# Zipf's law in China's local government work reports: A 21-year study using natural language processing and regression analysis

**Yanfang LI** [iD] *

School of International Languages, Xiamen University of Techonology, Xiamen, Fujian, China

* 656239885@qq.com

## Abstract

The examination and application of Zipf's law is a significant topic in quantitative linguistics. This study presents an in-depth empirical investigation of this law in 651 Chinese provincial government work reports (2003–2023). Employing natural language processing techniques (including Jieba word segmentation with a custom dictionary) and a double-logarithmic regression model, we analyzed word frequency distributions. Our findings indicate that the Zipf coefficient in these reports is close to 1, confirming general adherence to Zipf's law. Over the 21-year period, the Zipf coefficient exhibits fluctuations, with a notable inflection point in 2011, after which it follows a consistent upward trend. This shift is likely influenced by the 18th National Congress of the Communist Party of China, which marked a transition toward more standardized and centralized policy communication. While regional differences among eastern, central, western, and northeastern provinces are minimal, centrally governed municipalities exhibit higher Zipf coefficients than other provincial-level regions. Although our findings largely confirm the applicability of Zipf's Law to this specific corpus, this study is limited by the exclusion of prefecture- and county-level reports. Future research can address this limitation by incorporating a broader range of administrative levels and by conducting cross-country and cultural comparisons of political documents. Further investigation of alternate quantitative linguistic laws (e.g., Heaps' Law, Menzerath's Law) within this corpus is also warranted.

## 1. Introduction

In his seminal 1935 work, Psychobiology of Language—*An Introduction to Dynamic Philology* [1], George Kingsley Zipf examined word frequency distributions in natural language, introducing what is now recognized as Zipf's law. This principle asserts an inverse relationship between word frequency and its rank in a frequency table. Initially rooted in the "principle of least effort" [2], this principle has since been explored through the lens of information theory [3,4] and optimal coding [5], revealing a more

---

**Citation:** Li Y (2025) Zipf's law in China's local government work reports: A 21-year study using natural language processing and regression analysis. PLoS One 20(5): e0324713. https://doi.org/10.1371/journal.pone.0324713

**Data availability statement:** The data that support the findings are openly available in the [openICPSR] repository, at [https://doi.org/10.3886/E223761V1] To prevent any further errors in data uploading, all relevant data are also within the manuscript and its Supporting Information files.

**Funding:** The author(s) received no specific funding for this work.

**Competing interests:** The authors have declared that no competing interests exist.

nuanced understanding of its underlying mechanisms. Furthermore, deviations from a simple inverse relationship are frequently observed, revealing a two-regime behavior where high- and low-frequency words exhibit distinct scaling exponents [6]. This complexity extends beyond Zipf's law, where other significant linguistic regularities, such as Heaps' law, Menzerath's law, and the law of brevity (Zipf's law of abbreviation), also play significant roles in understanding language structure. Zipf's law of abbreviation posits an inverse relationship between word frequency and word length, illustrating how communicative efficiency drives linguistic evolution. Distinguishing these related but distinct linguistic regularities is essential, as the two Zipfian laws are sometimes conflated in the literature, leading to a lack of clarity in theoretical interpretation.

Importantly, Zipf's law transcends linguistics, serving as a fundamental characteristic of stochastic systems [7]. Its applicability spans diverse domains, including city size distribution [8], enterprise size distribution [9], and internet size distribution [10], affirming its relevance across social sciences and beyond. Zipfian distributions are regarded as "about as prevalent in social sciences as Gaussian distributions are in the natural sciences …, which implies that Zipf's Law captures a very fundamental regularity in the universe surrounding human beings" [11]. While extensive literature exists on Zipfian distributions [12], empirical investigations of Zipf's law in the context of Chinese local government work reports remain limited. This scarcity is noteworthy because these texts represent a distinctive example of highly formalized political communication. The specific characteristics of these documents, such as their structured format, specialized language, and emphasis on official policy, might result in deviations from Zipf's Law through the strategic use of particular terminology. However, this very need for concise and effective expressions could lead to a prioritization of common vocabulary and high-frequency words, potentially exhibiting the patterns predicted by Zipf's Law. Utilizing quantitative linguistic tools, particularly focusing on the analysis of word frequencies and their distributions, enables the identification of patterns and trends often obscured by other methods. This approach provides insights into the structural features of these texts and offers a means to objectively measure linguistic characteristics and identify deviations from expected patterns, thus revealing underlying dynamics within the data.

Through a large-scale analysis of a unique dataset comprising 651 reports from all 31 provincial-level administrative divisions (provinces, autonomous regions, and centrally governed municipalities) across a span of 21 years, this study explores the relationship between the formal characteristics of Chinese local government work reports and observed word frequencies. To do so we employ the double-logarithmic regression methodology of Torre et al. [13] within the theoretical framework of established linguistic principles and recent advancements in understanding Zipf's law.

This study provides empirical evidence regarding Zipf's Law within a large-scale corpus of Chinese local government work reports, demonstrating its general applicability while revealing significant inter-regional and inter-temporal variations in the adherence to the law's theoretical predictions. This research offers novel insights into the complexities of applying Zipf's Law to a formal political discourse.

## 2. Literature review

Zipf's law, a cornerstone of quantitative linguistics for nearly a century [1,2], has undergone significant theoretical expansion. Recent research offers more nuanced perspectives. Debowski provides a comprehensive framework integrating information theory and stochastic processes to model the emergence of Zipfian distributions [3], while Ferrer-i-Cancho et al. explore the connections between Zipf's law and optimal coding theory [5]. Kello et al. further broaden the application of scaling laws across the cognitive sciences, demonstrating their prevalence in various domains, notably in the organization of word frequencies [14]. Ferrer-i-Cancho argues that Zipf's Law's universality arises from fundamental organizational principles rather than specific linguistic features [15]. Additionally, empirical data frequently reveal a more complex two-regime behavior, characterized by distinct scaling exponents for high- and low-frequency words, which challenges to the simplistic inverse relationship [6].

The empirical evidence supporting Zipf's law spans diverse linguistic contexts. Studies have confirmed its presence in formulaic language [16] and spoken dialogue corpora [17]. However, the choice of analytical unit significantly impacts the observed adherence to this law, with Williams et al. demonstrating a stronger fit for phrases compared to individual words [18]. This highlights the need for nuanced models that account for various factors influencing word frequency distributions, as emphasized by Piantadosi [19]. Furthermore, Torre et al. offer a novel perspective by exploring the potential physical origins of linguistic patterns, including the relationship between lognormality and speech production [13]. Català et al. investigate the relationship between word frequency and semantic richness in Catalan, exploring the concept of "Zipf's laws of meaning." The findings contribute to understanding the interplay between lexical frequency and semantic complexity in natural language [20]. Further evidence of the prevalence of Zipf's law is also found in various Indian languages, Tibetan (a distinct language used in China), Korean, ancient Chinese texts, and Kazakh [21–25]. In a related study specific to Chinese texts, Xiao utilizes large, POS-tagged corpora, such as the GCCC and People's Daily corpus, revealing a general tendency toward Zipf's Law in Chinese newspaper and textbook texts, while noting some deviations [26]. Building upon this work, our study investigates whether similar patterns emerge in the distinct domain of formal political discourse, using local government work reports. Moreover, we refine the methodology by employing a custom dictionary for enhanced word segmentation and by explicitly considering regional and temporal variability in word frequencies—aspects that have not been addressed in prior research. Furthermore, this study expands our knowledge by assessing whether Zipf's Law applies to a political discourse, rather than simply to a more general linguistic text.

The utility of Zipf's law extends beyond simple verification, finding application in diverse fields. For instance, it has proven valuable in speech act analysis and text processing [27,28]. While methodological choices, such as word form-to-stem conversion, influence the parameters of Zipf's law, particularly affecting low-frequency words [28], the overall pattern of Zipfian distribution remains robust.

Beyond Zipf's law, several related linguistic laws influence language structure and usage. Heaps' law describes vocabulary growth with text length [29]; meanwhile, Menzerath's law links constituent length to the length of the encompassing structure [30]. The law of brevity highlights a preference for shorter linguistic units [2]. These laws, along with Zipf's law, offer a framework for comprehending the statistical regularities inherent in language. The interplay and potential interdependencies between these laws [31] and the broader implications across biological domains [32] represent an important area for future research. Furthermore, Català et al. extend the meaning-frequency law, revealing distinct patterns for content and function words [33], building upon previous findings on the relationship between word frequency and semantic properties [20].

In summary, although Zipf's Law was previously understood as mathematically simple, it has proven robust across diverse linguistic contexts. Recent research has illuminated its theoretical foundations and highlighted the importance of analytical unit selection. Related linguistic laws, such as Heaps', Menzerath's, and the law of brevity, along with emerging findings on semantic complexity, offer a richer understanding of language's statistical regularities. Further investigation into the interrelationships of these laws and their practical applications remains a crucial area of study.

   

This study contributes to the above-mentioned body of knowledge by examining Zipf's Law within the unique context of Chinese local government work reports, encompassing a dataset of 651 reports spanning 21 years from all 31 provincial-level administrative divisions. While previous research confirms the broad applicability of Zipf's Law, its specific relevance to this highly structured political discourse, and its inter-regional and inter-temporal variations, remain largely unexplored.

## 3. Methods and data

### 3.1 Data sources

This study compiles and organizes a dataset of 651 government work reports from 31 provincial-level administrative divisions in China, covering the years 2003–2023. This includes 22 provinces, five autonomous regions, and four centrally governed municipalities: Hebei, Shanxi, Liaoning, Jilin, Heilongjiang, Jiangsu, Zhejiang, Jiangxi, Anhui, Fujian, Shandong, Henan, Hubei, Hunan, Guangdong, Hainan, Sichuan, Guizhou, Yunnan, Shaanxi, Gansu, and Qinghai Provinces; Inner Mongolia, Ningxia Hui, Guangxi Zhuang, Xinjiang Uygur, and Tibet Autonomous Regions; Beijing, Tianjin, Shanghai, and Chongqing Municipalities. Noticeably, the government work reports included in this corpus are all written in Chinese. The reports are transcribed into a uniform text format, creating a dedicated corpus for analysis. Source material is obtained from the official websites of the respective provincial governments through a combination of web scraping techniques and manual compilation. To enhance processing efficiency, the encoding of the 651 text files is standardized, establishing the foundational corpus.

Data preprocessing is performed on each report using regular expressions and the Jieba word segmentation tool, after which the length of each report is calculated. Table 1 presents descriptive statistics of government work report lengths, including the number of observations (one report per year for 21 years), mean, standard deviation, and minimum and maximum values for each provincial-level administrative division. Analysis reveals that these government work reports are relatively short texts. These characteristics support the appropriateness of employing a double-logarithmic model in the empirical analysis [6].

During corpus construction, natural language processing (NLP) technology is applied to systematically process the data. The methodology includes the following steps:

(1) The documents are read and preliminarily processed using Python and core libraries such as re, collections, and Jieba:

- Regular expression functions from the re library are employed to eliminate punctuation, special symbols, and English characters (except for commonly used acronyms such as GDP and CPI), and Arabic numerals, ensuring data consistency and accuracy.

- The Jieba library's Chinese word segmentation function is used to segment the processed text, optimized for texts akin to government work reports. Jieba (https://pypi.org/project/jieba/), a widely used Chinese word segmentation library, offers several modes (precise, full, search engine). This study utilizes the precise mode, which is optimal for text analysis. While Jieba's default dictionary contains 300,000 entries, government work reports often include specialized terminology, political jargon, leadership speeches, and region-specific vocabulary. The use of the default dictionary alone could compromise the effectiveness of word segmentation. Therefore, it is necessary to construct a custom dictionary to supplement the entries and enhance the accuracy of the segmentation process. For instance, the phrase "Scientific Development Concept" (in Chinese pinyin: "ke xue fa zhan guan") might be segmented into two separate words ("kexue" and "fazhanguan") by default, but the custom dictionary optimizes this into a single token. Common acronyms in government work reports, such as the one found in the 2023 Hainan Provincial

**Table 1. Descriptive statistics on the length (total word count) of government work reports.**

| Provinces, autonomous regions, and cities | Obs | Mean | Std. dev. | Min* | Max* |
|---|---|---|---|---|---|
| Beijing Municipality | 21 | 17497.57 | 2407.587 | 12161 (2022) | 23017 (2020) |
| Tianjin Municipality | 21 | 16623.52 | 2293.059 | 12407 (2015) | 22616 (2023) |
| Hebei Province | 21 | 18526 | 3449.536 | 6297 (2016) | 23241 (2006) |
| Shandong Province | 21 | 17876.05 | 2706.407 | 12696 (2014) | 22451 (2003) |
| Jiangsu Province | 21 | 17206.62 | 2096.263 | 14467 (2004) | 21687 (2023) |
| Shanghai Municipality | 21 | 19139.38 | 3829.502 | 14145 (2014) | 26830 (2011) |
| Zhejiang Province | 21 | 16135.52 | 2598.072 | 11031 (2014) | 21671 (2023) |
| Fujian Province | 21 | 17023.05 | 4321.936 | 4995 (2013) | 27266 (2023) |
| Guangdong Province | 21 | 21988.71 | 1906.98 | 18315 (2003) | 26334 (2023) |
| Hainan Province | 21 | 17213.14 | 1897.117 | 14754 (2022) | 21720 (2016) |
| Shanxi Province | 21 | 17950.1 | 4465.355 | 5951 (2004) | 26155 (2023) |
| Henan Province | 21 | 18953.62 | 3593.8 | 11897 (2023) | 27345 (2021) |
| Anhui Province | 21 | 17862.14 | 3194.25 | 14108 (2006) | 24048 (2011) |
| Hubei Province | 21 | 18055.33 | 2508.075 | 13608 (2007) | 22124 (2016) |
| Hunan Province | 21 | 17875.24 | 2427.679 | 13434 (2014) | 21916 (2010) |
| Jiangxi Province | 21 | 16947.57 | 2377.403 | 13109 (2003) | 21777 (2023) |
| Heilongjiang Province | 21 | 16394.29 | 2291.836 | 12505 (2014) | 21650 (2010) |
| Jilin Province | 21 | 16025.86 | 4401.932 | 8555 (2005) | 26831 (2023) |
| Liaoning Province | 21 | 14848.95 | 2902.444 | 11418 (2013) | 19428 (2023) |
| Inner Mongolia Autonomous Region | 21 | 15569.86 | 2275.79 | 11524 (2020) | 19687 (2016) |
| Ningxia Hui Autonomous Region | 21 | 17428.62 | 2033.535 | 12269 (2022) | 22735 (2023) |
| Shaanxi Province | 21 | 16482.57 | 2305.805 | 11841 (2020) | 21491 (2008) |
| Gansu Province | 21 | 18626.38 | 2195.785 | 13141 (2004) | 21780 (2015) |
| Sichuan Province | 21 | 17286 | 2651.493 | 13569 (2004) | 24154 (2023) |
| Chongqing Municipality | 21 | 20138.81 | 2427.048 | 17476 (2018) | 25905 (2023) |
| Guizhou Province | 21 | 18641.05 | 2465.253 | 15246 (2012) | 23802 (2006) |
| Guangxi Zhuang Autonomous Region | 21 | 19907.19 | 2108.857 | 15955 (2021) | 23682 (2013) |
| Yunnan Province | 21 | 17282.81 | 3217.106 | 7039 (2007) | 21486 (2008) |
| Qinghai Province | 21 | 16376.43 | 1796.896 | 12508 (2013) | 19180 (2016) |
| Xinjiang Uygur Autonomous Region | 21 | 19749.95 | 5282.118 | 7383 (2009) | 31552 (2023) |
| Tibet Autonomous Region | 21 | 16441.86 | 3781.212 | 8654 (2006) | 23002 (2011) |

NOTE:1.() indicates the year in which the minimum and maximum values occur 2.The length of each report is measured as the total word count, including repeated words, after preprocessing the text.

Government Work Report (in Chinese pinyin "yi ben san ji si liang ba zhu") would be segmented into three separate tokens ("yiben", "sanjisiliang" and "bazhu") by default, whereas the custom dictionary consolidates it into a single token.

• Word frequencies are then calculated using Python's Counter package and ranked in descending order, yielding two variables: word frequency ($S$) and rank order ($R = 1, 2, \ldots$).

(2) With 651 documents, these steps are executed individually for each document, resulting in 651 distinct processing operations. This rigorous approach ensures the high-quality construction of the corpus and provides a solid foundation for subsequent NLP analyses.

## 3.2. Empirical model

Zipf's law is satisfied when the absolute frequency (S) of a word and its rank order (R) adhere to the following power-law relationship:

$$S \propto \frac{1}{R^{\beta}},$$

Where $\beta$ is the Zipf coefficient. This relationship can be expressed equivalently as:

$$S = \frac{c}{R^{\beta}} = cR^{-\beta}.$$

Taking the logarithm of both sides transforms the relationship as follows for analytical purposes:

$$\log S = \log c - \beta \log R.$$

If $\beta \approx 1$, the data exhibit adherence to Zipf's law, indicating a typical rank-size distribution of word frequencies.

If $\beta > 1$, the distribution is more skewed, suggesting that a few high-frequency words dominate the text.

If $\beta < 1$, the distribution is flatter, implying a more even distribution of word frequencies across the vocabulary.

When $\beta = 1$, the law simplifies to:

$$S \times R = c,$$

where $c$ is a constant. This means the product of a word's rank and frequency is constant: the most frequent word ($R = 1$) is twice as frequent as the second ($R = 2$), the third is three times less frequent than the first, and so on. Verification of Zipf's law involves assessing whether $\beta$ is approximately 1. Significant deviations indicate non-compliance, while values near 1 suggest adherence.

To empirically verify Zipf's law, we use regression analysis. Specifically, we use ordinary least squares to estimate the coefficients. Following Torre et al.'s [13] methodology, we estimate a double-logarithmic regression of the form:

$$\log S_{it} = \log\ c - \beta \log R_{it} + \varepsilon,$$

where $i$ represents the provincial-level administrative division, $i = 1, 2 \ldots 31$; $t$ represents year, $t = 2003,\ 2004, \ldots,\ 2023$, and $\varepsilon$ denotes the random error term.

Stata is widely used in econometrics and statistical analysis across various disciplines, including economics, management, and linguistics. Stata 16.0 is thus employed to conduct regression analysis on word frequency and rank order from government work reports over 21 years. This yields the Zipf coefficient and enables verification of Zipf's law for these reports. While Corral et al. examine both word forms and lemmas across multiple languages, observing Zipfian distributions in both, this study focuses its analysis exclusively on word forms [28]. This decision stems from the fact that the lemmatization, a process essential for morphologically rich languages like English, is not required for the Chinese language present in these government work reports due to its relatively straightforward morphology. The technical roadmap for this article is illustrated in Fig 1.

## 4. Empirical results and discussion

### 4.1. The Zipf coefficients close to 1

Regression analysis yields 651 Zipf coefficients ($\beta$ values, the regression coefficient is negative $\beta$), all exhibiting p-values of 0, indicating statistical significance at $p < 0.001$. In our investigation of the correlation between Zipf's Law and the

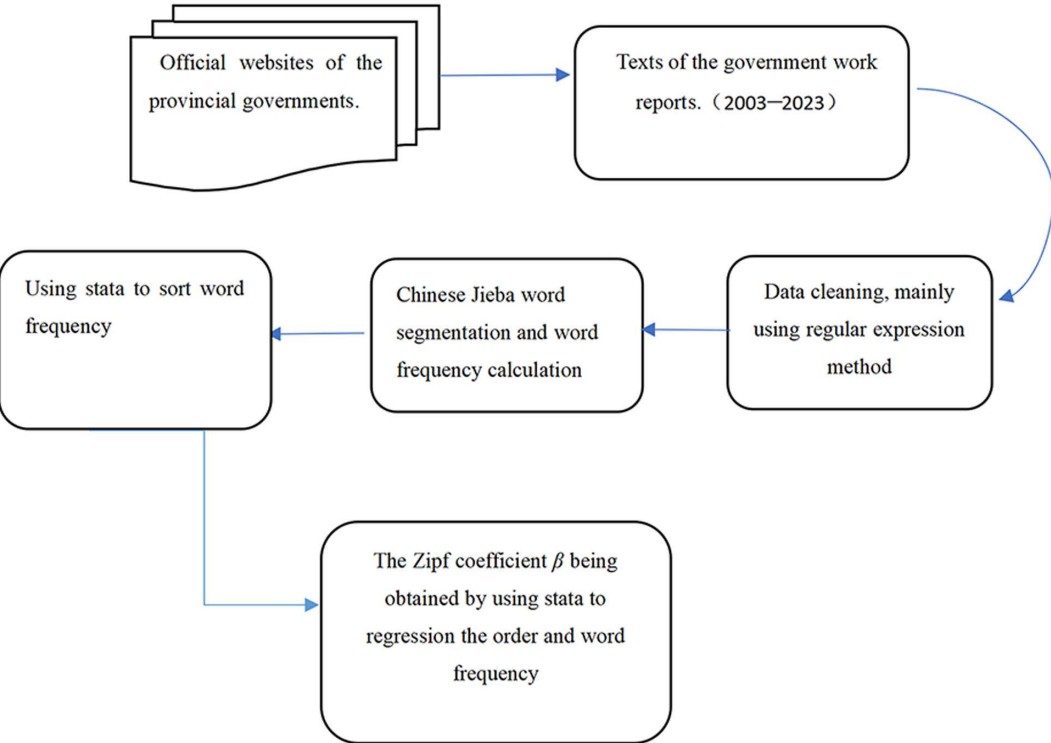

**Fig 1. Technical roadmap.**

content of local government work reports, we employ regression analysis using the empirical model facilitated by Stata 16. The findings reveal that the regression models exhibit consistently high $r^2$ values (all > 0.8), consistent with previous research by Piantadosi [19] and Mehri and Jamaati [34].

The calculated Zipf coefficients ($\beta$ values) achieve high statistical significance, with all associated $p$-values below 0.001. This significance underscores the pivotal role and accuracy of these coefficients in revealing distributional attributes within local government work reports, including policy emphases, economic metrics, and social welfare concerns, along with their fluctuations across various years and jurisdictions.

Statistical analysis indicates an average Zipf coefficient of approximately 0.819, which, while close to 1, exhibits a slight divergence. It is important to note that the theoretical value for a perfect Zipfian distribution is $\beta = 1$, a value rarely observed in real-world data. Our results demonstrate a general tendency towards this ideal, indicating a strong degree of content concentration within the reports, in which a few common words occur with much higher frequency than the others, which are less common. The coefficient range, from a nadir of 0.668 to an apex of 0.925, suggests subtle variations in thematic architecture among different local governments or periods; however, these variations do not significantly disrupt the overarching pattern of centralized distribution. Additionally, the standard deviation of 0.029 highlights the uniformity of the Zipf coefficient.

The distribution of these coefficients is visually represented in Fig 2, illustrating that most coefficients cluster near 1. This graphical representation provides compelling empirical support for Zipf's Law and the discernible stylistic and content preferences in Chinese local government work reports.

While Zipf's law theoretically predicts a $\beta$ value of 1, empirical studies show variations in $\beta$ between spoken and written language [35]. In spoken language, $\beta$ is influenced by factors such as the speaker's age, whereas in written language,

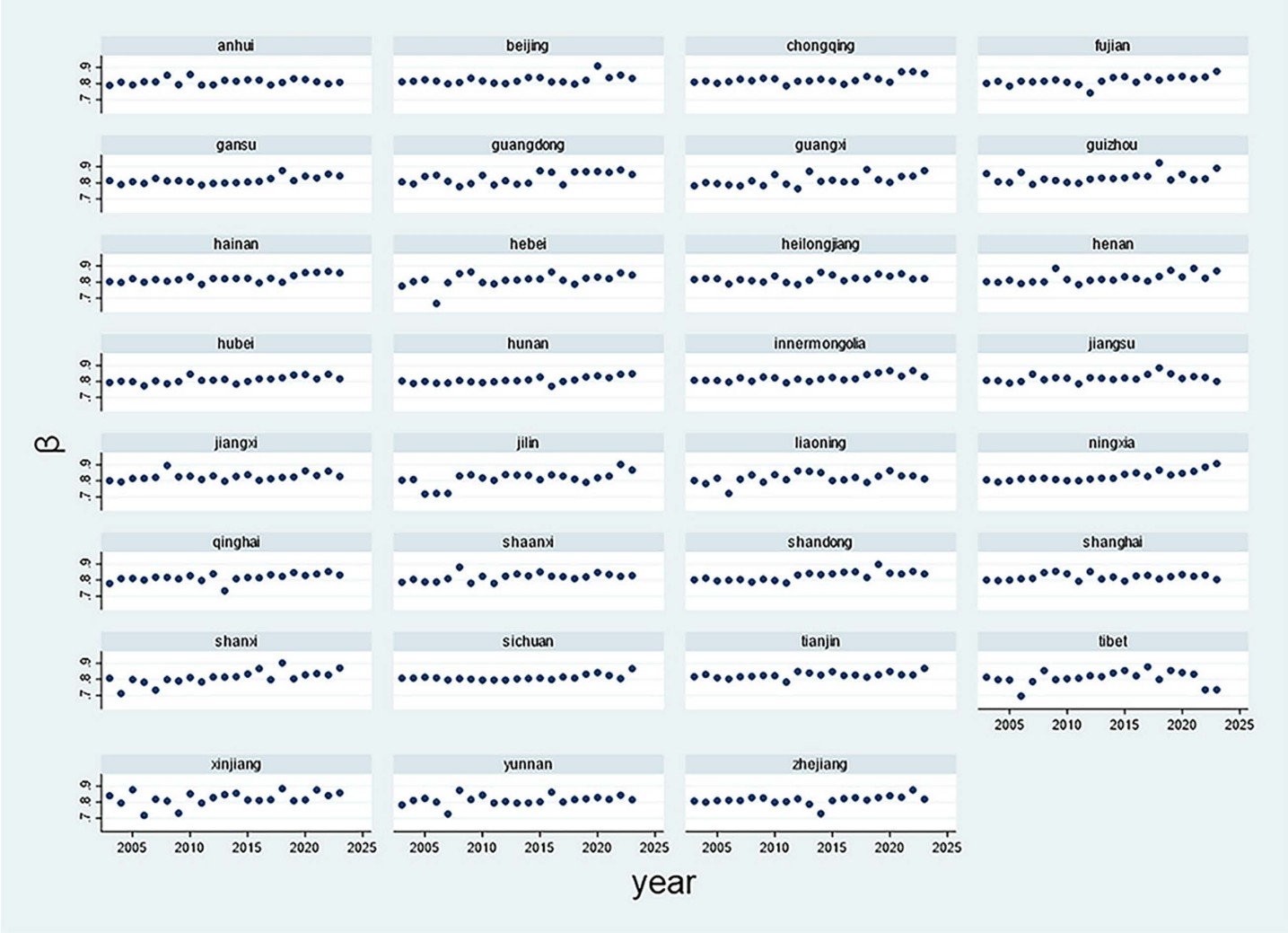

**Fig 2. Temporal trends of Zipf coefficients by all provincial-level regions (2003–2023).**

text length is a significant determinant. The $\beta$ value reflects the balance between communicative efficiency and the cost of communication [36]; higher $\beta$ values indicate a greater emphasis on communicative efficiency. The observed $\beta$ values in the government work reports suggest a prioritization of this efficiency.

### 4.2. The heterogeneity of Zipf coefficients

Further examination of the Zipf coefficient distribution reveals that only 1% of the coefficients fall below 0.722, while 10% are below 0.791. With all $\beta$ values collectively below 1 (indicating a flatter, more even distribution of word frequencies, as defined in Section 3.2), these results demonstrate that word frequency distributions in local government work reports exhibit relative balance, with minimal skewness toward high-frequency dominance.

Specifically, over 90% of the Zipf coefficients exceed 0.79, indicating significant content concentration within most reports. This reflects the substantial emphasis local governments place on key issues addressed in their reports. Additionally, more than 75% of the coefficients exceed 0.803, further underscoring this trend and suggesting that local

governments prioritize a limited number of critical topics. Notably, only 1% of the Zipf coefficients exceed 0.895, suggesting that the reports generally exhibit characteristics consistent with Zipf's law, although some exhibit notable deviations in content distribution. Such deviations may result from region-specific policy requirements, socioeconomic conditions, or other external factors.

The data suggest that when a Zipf coefficient exceeds 0.8, approaching the ideal value of 1, approximately 10% of the content in China's local government work reports deviates from Zipf's law. This deviation may warrant further investigation to understand its causes and implications. While the maximum observed $\beta$ value reached 0.925, these figures do not necessarily indicate non-compliance with Zipf's Law. Instead, this systematic deviation underscores the combined influence of distinct political dynamics, regional nuances, and the inherent structural constraints of government reports. The distribution of Zipf coefficients is depicted in Fig 3.

A detailed examination of the data (Table 2) reveals significant variation among provincial-level regions. These regional differences are systematically summarized in Table 2, which provides descriptive statistics (mean, standard deviation, and minimum and maximum values) of Zipf coefficients across all 31 provincial-level regions. Over the 21-year period, Guizhou Province consistently demonstrates the highest average Zipf coefficient, with a mean of 0.833. Conversely, Hunan Province records the lowest average, with a mean of 0.809. Guizhou Province also achieves the highest peak Zipf coefficient at 0.925, while Hubei Province registered the lowest peak value at 0.846. Regarding minimum coefficients, Beijing has the highest at 0.797, whereas Hebei Province recorded the lowest at 0.668. These findings underscore the notable heterogeneity in Zipf coefficients among local government work reports from various regions, suggesting subtle differences in their thematic structure and linguistic patterns, thereby illuminating previously unexamined regional distinctions in Chinese political discourse. Despite this variability, the mean of the 31 provincial-level Zipf coefficients is 0.819, with a minimum of 0.809, a maximum of 0.833, and a standard deviation of 0.007. This indicates a high degree of stability with no statistically significant differences across provincial-level regions and centrally governed municipalities. This consistency may be attributed to the highly formalized and standardized nature of Chinese local government work reports.

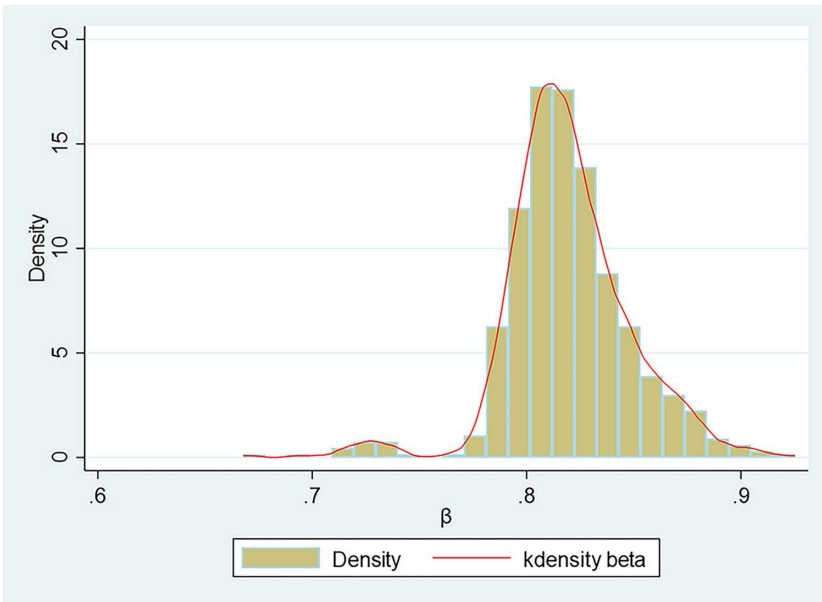

**Fig 3. Histogram and kernel density estimation of Zipf coefficients.**

**Table 2. Descriptive statistics on Zipf coefficients of the provincial-level regions.**

| Provinces, autonomous regions, and cities | Mean | Std. dev. | Min* | Max* |
|---|---|---|---|---|
| Beijing Municipality | 0.824 | 0.025 | 0.797 (2018) | 0.910 (2020) |
| Tianjin Municipality | 0.827 | 0.018 | 0.783 (2011) | 0.868 (2023) |
| Hebei Province | 0.813 | 0.042 | 0.668 (2009) | 0.864 (2006) |
| Shandong Province | 0.825 | 0.028 | 0.782 (2011) | 0.898 (2019) |
| Jiangsu Province | 0.821 | 0.022 | 0.786 (2011) | 0.885 (2018) |
| Shanghai Municipality | 0.819 | 0.019 | 0.793 (2011) | 0.855 (2009) |
| Zhejiang Province | 0.814 | 0.027 | 0.729 (2014) | 0.876 (2022) |
| Fujian Province | 0.820 | 0.027 | 0.742 (2012) | 0.877 (2023) |
| Guangdong Province | 0.831 | 0.036 | 0.777 (2008) | 0.881 (2022) |
| Hainan Province | 0.823 | 0.023 | 0.787 (2011) | 0.867 ( (2022) |
| Shanxi Province | 0.811 | 0.042 | 0.712 (2004) | 0.903 (2018) |
| Henan Province | 0.825 | 0.030 | 0.786 (2011) | 0.887 (2009) |
| Anhui Province | 0.812 | 0.020 | 0.789 (2003) | 0.857 (2010) |
| Hubei Province | 0.811 | 0.020 | 0.773 (2006) | 0.846 (2010) |
| Hunan Province | 0.809 | 0.020 | 0.771 (2016) | 0.848 (2023) |
| Jiangxi Province | 0.826 | 0.024 | 0.794 (2004) | 0.895 (2008) |
| Heilongjiang Province | 0.822 | 0.020 | 0.786 (2012) | 0.862 (2014) |
| Jilin Province | 0.813 | 0.045 | 0.719 (2005) | 0.902 (2022) |
| Liaoning Province | 0.817 | 0.032 | 0.722 (2006) | 0.864 (2020) |
| Inner Mongolia Autonomous Region | 0.822 | 0.021 | 0.792 (2011) | 0.868 (2022) |
| Ningxia Hui Autonomous Region | 0.829 | 0.030 | 0.792 (2014) | 0.907 (2023) |
| Shaanxi Province | 0.819 | 0.025 | 0.780 (2011) | 0.881 (2008) |
| Gansu Province | 0.817 | 0.023 | 0.787 (2011) | 0.877 (2018) |
| Sichuan Province | 0.812 | 0.017 | 0.796 (2012) | 0.867 (2023) |
| Chongqing Municipality | 0.825 | 0.023 | 0.786 (2011) | 0.875 (2022) |
| Guizhou Province | 0.833 | 0.032 | 0.791 (2007) | 0.925 (2018) |
| Guangxi Zhuang Autonomous Region | 0.816 | 0.033 | 0.764 (2012) | 0.884 (2018) |
| Yunnan Province | 0.813 | 0.030 | 0.727 (2007) | 0.874 (2008) |
| Qinghai Province | 0.816 | 0.026 | 0.734 (2013) | 0.853 (2022) |
| Xinjiang Uygur Autonomous Region | 0.823 | 0.042 | 0.718 (2006) | 0.884 (2018) |
| Tibet Autonomous Region | 0.809 | 0.044 | 0.697 (2006) | 0.879 (2017) |

NOTE: () indicates the year in which the minimum and maximum values occur.

These reports are unique documents that articulate the policy directives, developmental goals, and administrative strategies of local governments. They are written in a formal and authoritative tone, reflecting the official stance and commitments of the government. They are also known for their structured format; they typically begin with a review of the previous year's achievements and challenges, followed by an outline of the goals and plans for the upcoming year. They also include detailed statistical data to substantiate the government's claims and objectives. We acknowledge that the features of government work reports, owing to their standardized format, are not readily attributable to individual author-ship, a factor often associated with stylistic variation. Moreover, in Zipf's Law, $\beta$ theoretically equals 1, although empirical investigations show deviations influenced by text length, as evidenced by Shakespeare's longer texts with a higher $\beta$, compared to the shorter reports in our study. However, our results also illustrate that text length is not the only factor influencing the Zipf coefficients. These findings demonstrate that in the context of Chinese local government work reports,

word frequency distributions do indeed provide information regarding regional variations in stylistic choices and thematic priorities.

Further analysis reveals that only four provincial-level regions exhibited minimum Zipf coefficients after 2012, with the remaining 27 displaying minima prior to 2012. This may be attributed to the 18th National Congress of the Communist Party of China in 2012, marking the commencement of a new era.

Furthermore, minima frequently coincide with the year of or preceding the Party Congress; for instance, 14 provincial-level regions show minima in 2011 or 2012. Conversely, maxima predominantly occurred post-2012 (23 provincial-level regions), with 13 instances after 2020. However, influences beyond Party Congresses likely affect the Zipf coefficient, including regional characteristics, year-specific factors, individual official characteristics, and the use of political language in the reports. Comparing report length and Zipf coefficient extrema, only four provincial-level regions exhibit alignment between the years of minimum length and minimum Zipf values, and six show alignment between maximum length and maximum Zipf values. In the remaining cases, no such correspondence is observed.

### 4.3. Temporal trend of Zipf coefficients

To provide a clearer illustration of the variations in Zipf coefficients, the analysis presents Figs 4 and 5.

Fig 4 illustrates the annual mean value of the Zipf coefficient for all provincial-level regions between 2003 and 2023, revealing a distinct trajectory that can be divided into three phases. The period prior to 2011 is marked by significant fluctuations, with notable peaks observed in 2008 and 2010, followed by a substantial decline to a low of 0.794 in 2011. After 2011, the Zipf coefficient exhibits a steady upward trend, accelerating after 2017 and reaching values above 0.84 between 2021 and 2023. This pattern may reflect a growing emphasis on the strategic use of high-frequency words in government reports, potentially driven by the need for clearer and more standardized communication in policy discourse after 2011.

Fig 5 provides a more granular analysis by segmenting the Zipf coefficients into quintiles, illuminating the heterogeneity underlying the aggregate trend. The visualization confirms 2011 as a critical inflection point across all quintile groups, with a synchronous decline followed by a differentiated but universally positive trajectory thereafter. While maintaining their

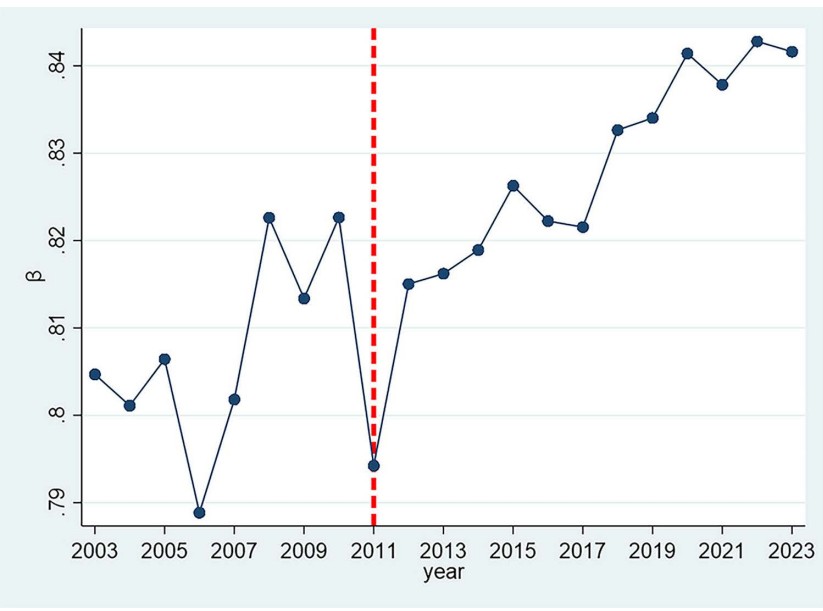

**Fig 4. The evolution of the annual average value of Zipf coefficients(2003–2023).**

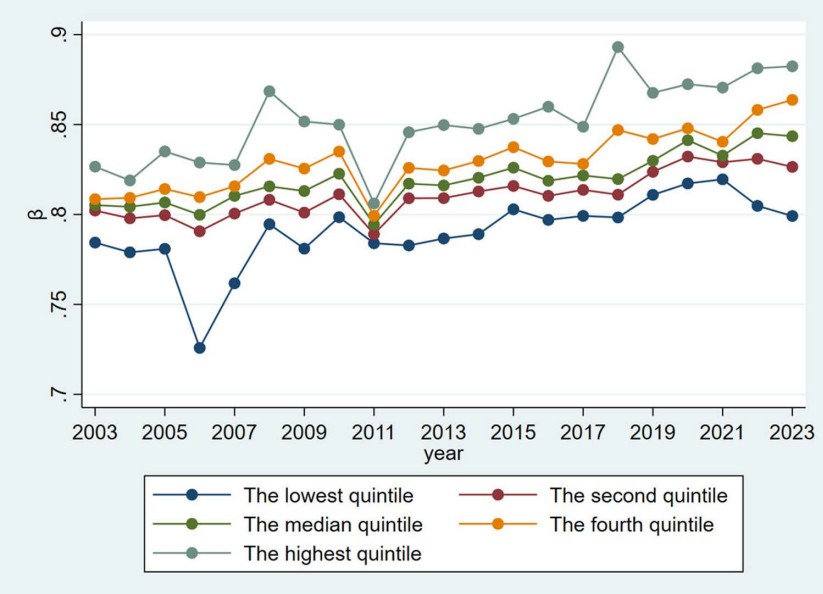

**Fig 5. Zipf Coefficient Distribution Across Quintiles (2003–2023).**

relative positions, all quintiles show varying recovery rates post-2011. The highest quintile demonstrates the most significant increase, peaking at approximately 0.89 in 2018 before stabilizing between 0.86 and 0.88 in subsequent years. The lowest quintile, though initially exhibiting subdued values, mirrors this pattern with a delayed but notable recovery after 2017. The synchronized quintile movements, despite their magnitude differences, suggest systemic influences transcending regional particularities.

This shift coincides with the 18th National Congress of the Communist Party of China in 2012, which ushered in a new era. The pre-2011 period, marked by fluctuating coefficients, may reflect greater diversity in policy language and themes prior to the Congress. In contrast, the post-2011 trend toward higher Zipf values suggests a more concentrated vocabulary distribution, likely resulting from increased standardization in political discourse as well as a more centralized approach to policy communication.

### 4.4. The difference of Zipf coefficients across regions

To better understand the linguistic patterns in China's government work reports, it is crucial to examine how the distribution of the Zipf coefficient varies across different regions and administrative categories. Fig 6 illustrates the temporal distribution of the Zipf coefficient for China's four major regions: Northeast (Liaoning, Jilin, Heilongjiang), East (Beijing, Tianjin, Hebei, Shandong, Jiangsu, Shanghai, Zhejiang, Fujian, Guangdong, Hainan), Central (Shanxi, Henan, Anhui, Hubei, Jiangxi, Hunan), and West (Chongqing, Sichuan, Shaanxi, Yunnan, Guizhou, Guangxi, Gansu, Qinghai, Ningxia, Tibet, Xinjiang, Inner Mongolia). The data reveal significant fluctuations prior to 2011, including a decline in the Northeast region in 2006, followed by a recovery in subsequent years. A synchronous decline around 2011 is observed across all regions, reinforcing 2011 as a critical inflection point. After 2011, all regions display an upward trajectory, although with distinct patterns. The Eastern region exhibits the most pronounced increase after 2018, while the Northeastern region shows the greatest volatility overall. Despite these regional differences, the general similarity in trends suggests that linguistic patterns in government work reports are more influenced by national-level policy directives than by regional distinctions.

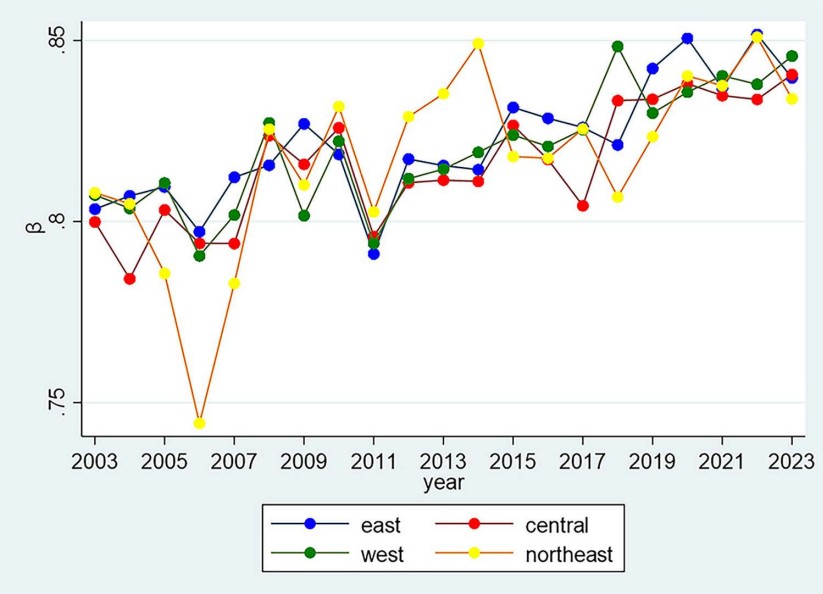

**Fig 6. Zipf Coefficient Variations Across China's Four Regions (2003–2023).**

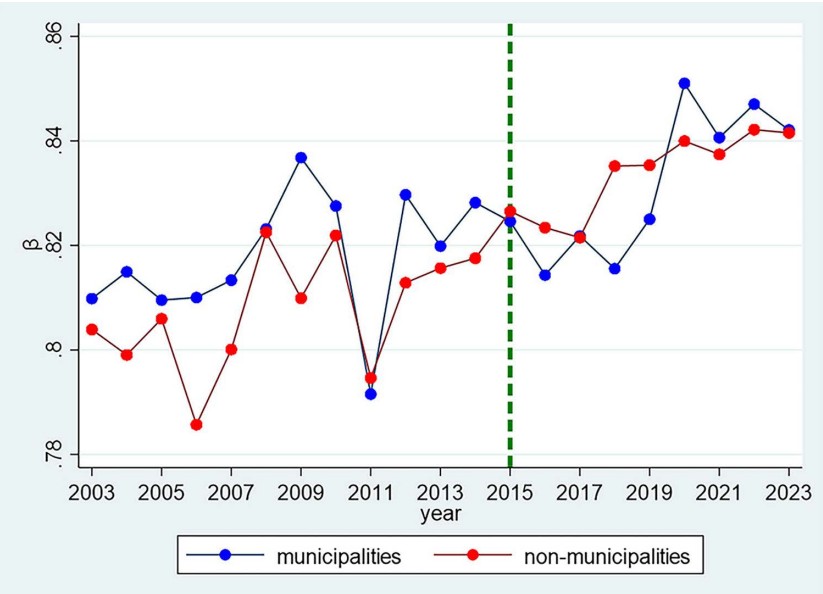

**Fig 7. Temporal variation of Zipf coefficients for municipalities and other provincial-level regions (2003–2023).**

Building on this regional analysis, Fig 7 divides provincial-level regions into two categories: centrally governed municipalities (Beijing, Tianjin, Shanghai, and Chongqing) and the remaining provincial-level regions. The figure reveals that the Zipf coefficient trends for both groups are similar, with centrally governed municipalities consistently exhibiting higher coefficients than non-municipalities during the periods 2003–2015 and 2020–2023. Only in five years (2011, 2015, 2016,

2018, and 2019) did centrally governed municipalities exhibit lower coefficients than other provincial-level regions. This pattern may be attributed to the unique role of centrally governed municipalities as political, economic, and cultural hubs, which likely leads to greater centralization and uniformity in policy formulation and implementation. The higher Zipf coefficients in centrally governed municipalities suggest a more concentrated vocabulary distribution, likely driven by their advanced administrative systems and specialized policy vocabularies. In contrast, non-municipalities may display lower coefficients due to greater policy diversity and regional development imbalances, reflecting the socio-economic disparities across provincial-level regions. The elevated coefficients in centrally governed municipalities during specific periods (e.g., 2003–2015 and 2020–2023) may also signal their closer alignment with national strategic priorities, such as economic reforms or major political events, which could have influenced the linguistic structure of their reports. Overall, these findings underscore the complex interaction between administrative divisions, regional characteristics, and temporal factors in shaping the linguistic patterns of government work reports.

## 5. Conclusion and forward direction

Zipf's law is arguably "the most well-known statement of quantitative linguistics" and is widely recognized in the broader study of complex systems [37,38]. Although the existing literature on Zipf's law is extensive, as far as we know, our study represents the first large-scale analysis of Zipf's law in China's government work reports, significantly expanding existing quantitative linguistic research and contributing to the political literature. Moreover, by focusing on formalized political discourse, our study provides a novel dataset that demonstrates how formal political documents align with Zipf's Law. As such, we are uniquely positioned to provide a substantiated assessment of the law's validity in this specific context.

Employing a corpus research methodology, this study compiles a comprehensive dataset of government work reports from all 31 provincial-level administrative regions in China, spanning 21 years. The Python programming language is utilized for natural language processing, leveraging the Jieba word segmentation library with a custom dictionary to ensure precise segmentation. Following this, a statistical analysis of word frequency is conducted. An empirical model is then developed to test Zipf's law, with regression analysis performed using Stata 16.0 to determine the Zipf coefficient. The results indicate that Chinese local government work reports generally conform to Zipf's law, but exhibit notable variability in Zipf coefficients across different years, though not significantly across geographical regions.

The temporal analysis reveals a clear trend in Zipf coefficients, with a notable inflection point in 2011 that coincides with the 18th National Congress of the Communist Party of China in 2012. Prior to this period, the coefficients demonstrated considerable fluctuations. However, post-2011, they exhibited a consistent upward trajectory, suggesting a trend toward greater standardization in political discourse. Our quintile analysis (Fig 5) further reinforces this finding, showing distinct patterns in the distribution of Zipf coefficients across different percentile groups over time. Despite these temporal trends, our regional comparison across Eastern, Central, Western, and Northeastern regions of China reveals remarkably similar patterns, indicating that geographical factors do not significantly influence the linguistic structures of government work reports. This suggests that the observed changes in linguistic patterns are more likely driven by nationwide policy shifts rather than regional characteristics. Furthermore, centrally governed municipalities consistently display higher coefficients than other provincial-level regions throughout most of the examined period. This disparity may reflect the status of central municipalities as political and economic hubs, where policy formulation tends to be more centralized.

Rather than simply validating Zipf's law, our most significant contribution lies in demonstrating the importance of examining both temporal trends and different administrative divisions (particularly the distinction between centrally governed municipalities and other provincial-level regions) to achieve a comprehensive understanding of linguistic patterns within formal political discourse in China. Moreover, utilizing a quantitative linguistic perspective, our analysis reveals a relationship between the structural attributes of political documents and their word frequency patterns, which illustrates the potential for further studies of other political documents. In doing so, this work also shows how quantitative linguistic techniques can be employed to investigate the construction of these documents, offering insights into their structure and

subtle variations across time and regions. The application of natural language processing tools facilitates the efficient analysis of large-scale corpus data, laying a critical foundation for more rigorous scientific inquiry. This study represents a pioneering empirical investigation of Zipf's law within the context of Chinese local government work reports, providing important insights into their linguistic properties. It also demonstrates how natural language processing and regression analysis reveal patterns in political communication. The core methods and theoretical frameworks employed—natural language processing and double-logarithmic regression—can be adapted and complemented for use in analyzing other discursive contexts, while the Jieba library with the custom dictionary is specific to Chinese text.

In linguistics, the search for Zipf's law has traditionally been conducted on extremely limited text sets [39]. This study aligns with that precedent by focusing on provincial government work reports. However, it is important to note that China has about 650 cities and over 1,300 county-level administrative units, which provide richer and more detailed information than the sample of provincial ones. Furthermore, the term "Chinese language" in this study is specifically defined as Standard Mandarin, the official language of the People's Republic of China. Nevertheless, due to resource constraints and challenges in data collection, this study could not include prefecture- and county-level government work reports, potentially affecting the comprehensiveness and representativeness of the findings.

Future research should aim to expand the textual scope by incorporating prefecture- and county-level administrative reports and conducting a comparative cross-country and cultural analysis with political documents to further explore the applicability of Zipf's Law. Furthermore, as this study is limited to an examination of Chinese government work reports, further research can also employ other quantitative linguistic laws, such as Zipf's meaning-frequency law, Heaps' Law, and Menzerath's Law.

## Supporting information

**S1 File. Data and code of government reports for Zipf's law.**
(ZIP)

## Acknowledgments

I would like to express my sincerest gratitude to all those who have contributed to the completion of this paper. First, my deepest gratitude goes to the anonymous reviewers for their insightful comments and constructive feedback, which have greatly contributed to the improvement of this manuscript. I also wish to extend my heartfelt appreciation to Professor Shi Chunxu, who gave me advice and guidance through all stages of the writing.

## Author contributions

**Writing – original draft:** Yanfang Li.

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
