## [Decision Letter · Decision Letter 0]

3 Dec 2024

PONE-D-24-44352Zipf’s Law in China’s Local Government Work Reports: A 21-year Study Using Natural Language Processing and Regression AnalysisPLOS ONE

Dear Dr. Li,

Thank you for submitting your manuscript to PLOS ONE. After careful consideration, we feel that it has merit but does not fully meet PLOS ONE’s publication criteria as it currently stands. Therefore, we invite you to submit a revised version of the manuscript that addresses the points raised during the review process.

The manuscript has been evaluated by three reviewers, and their comments are available below.

The reviewers have raised a number of concerns that need attention. They request that additional quantifications are performed as well as further statistical analysis, and that the clarity of data representation could be improved. I also note that one reviewer has recommended that you cite specific previously published works. As always, I recommend that you please review and evaluate the requested works to determine whether they are relevant and should be cited. It is not a requirement to cite these works. 

Could you please revise the manuscript to carefully address the concerns raised?==============================

We look forward to receiving your revised manuscript.

Kind regards,

Johanna Pruller, Ph.D.

Staff Editor

PLOS ONE

2. Thank you for uploading your study's underlying data set. Unfortunately, the repository you have noted in your Data Availability statement does not qualify as an acceptable data repository according to PLOS's standards. At this time, please upload the minimal data set necessary to replicate your study's findings to a stable, public repository (such as figshare or Dryad) and provide us with the relevant URLs, DOIs, or accession numbers that may be used to access these data. For a list of recommended repositories and additional information on PLOS standards for data deposition, please see https://journals.plos.org/plosone/s/recommended-repositories.

Additional Editor Comments (if provided):

Reviewers' comments:

Reviewer's Responses to Questions

**Comments to the Author**

1. Is the manuscript technically sound, and do the data support the conclusions?

Reviewer #1: Partly

Reviewer #2: Partly

2. Has the statistical analysis been performed appropriately and rigorously? 

Reviewer #1: No

Reviewer #2: No

3. Have the authors made all data underlying the findings in their manuscript fully available?

Reviewer #1: Yes

Reviewer #2: Yes

4. Is the manuscript presented in an intelligible fashion and written in standard English?

Reviewer #1: Yes

Reviewer #2: Yes

5. Review Comments to the Author

Reviewer #1: The study presented in the document examines the application of **Zipf's Law** to local government work reports in China over a span of 21 years (2003–2023). Using natural language processing tools (such as Chinese word segmentation with the Jieba library) and regression analysis in Stata, the researchers analyzed a corpus of 651 reports from 31 provinces and regions.

The paper aims to demonstrate the confirmation of Zipf's Law

- The analyzed reports generally follow Zipf's Law, which establishes an inverse relationship between a word's frequency and its rank in a frequency table.

- Although variations in Pareto coefficients (key to measuring this relationship) exist, they tend to values close to 1, indicating linguistic patterns consistent with Zipf's Law.

However, quantitative results of analysis and inference are necessary to provide strong evidence for the presence of the Law. At a minimum, a descriptive analysis that helps readers understand the strength of the empirical evidence, as well as the specific deficiencies noted in the text, is required. In particular, the heterogeneity across regions:

- Some provinces showed significant differences in Pareto coefficients. For instance, Guizhou had the highest average, while Hunan recorded the lowest.

- These differences reflect variations in priority topics and linguistic styles due to regional or political factors.

A clear presentation of data visualizations would help readers frame the author’s claims within an empirical evidence context. In general, the methodological details are sparse. To strengthen the article's credibility, additional details enabling result replication would be beneficial. In summary, the paper mentions using Zipf’s Law to analyze predominant words and topics.

Summarizing, there are two main practical implications:

- The findings suggest that adhering to consistent linguistic patterns could improve the clarity and transparency of government reports.

- The results may also be useful for training officials in drafting reports and for comparing government communication styles across historical or international contexts.

This pioneering work combines quantitative linguistics and governance, proposing practical improvements for governmental communication in China based on linguistic principles. It is a highly interesting study, and I recommend its publication with major revisions.

Reviewer #2: REVIEW: Zipf’s Law in China’s Local Government Work Reports: A 21-Year Study Using Natural Language Processing and Regression Analysis

(PONE-D-24-44352)

In this study, the authors analyze the conformity of China’s local government work reports (LGWRs) with Zipf’s law over a 21-year period using natural language processing and regression analysis. They use Stata’s regression methods and Jieba word segmentation to investigate linguistic patterns in 651 LGWRs from 31 provinces, offering insights into the relationship between word frequency distributions and report structure. Their results suggest that LGWRs generally conform to Zipf’s law, though variation exists across time and regions. The authors propose that aligning linguistic patterns with Zipf’s law could enhance clarity and transparency in governmental communication.

The research has clear potential and provides a valuable application of quantitative linguistics to political documents. However, several critical areas of the work require improvement, particularly regarding the theoretical foundation, references to key literature, and the interpretation of Zipf’s law in this specific context. Below, I outline the main issues and offer suggestions for revisions.

After the review, I think the authors have very good data, but they have not been properly analyzed. I have found the approach very interesting, but unfortunately it needs a new statistical analysis of the data and review of the recent literature on Zipf’s law and other interesting Linguistic Laws, which may have been overlooked by the authors (not being strictly a topic in their approach), which is common in interdisciplinary work like this, on the other hand very necessary to do comparative studies in between social sciences and linguistics

Finally, I believe that the work requires a general review and a new analysis of the data. It can be published in Plos One, but only after some major revisions and considerations pointed below. I encourage the authors to make these changes in order to expand our knowledge of the common quantitative patterns in human language and communication systems. I congratulate authors on their hard experimental work, but accordingly to that effort they should improve their analysis and discussion, which will help to increase the scientific impact of their studies.

*** Main issues ***

Abstract

0.- Rewrite after review. The abstract is concise but lacks specificity about the theoretical implications, practical consequences and limitations of the study. The authors should clearly state whether their findings confirm or challenge existing hypotheses about Zipf’s law in structured political discourse. Additionally, the abstract could benefit from a brief mention of methodological limitations, such as potential biases in word segmentation and data preprocessing.

1.- Authors claim that (lines 15-18): “Our findings have practical implications for enhancing government report clarity and transparency by adhering to linguistic patterns that align with Zipf’s law. This study offers a valuable framework for gaining deeper insights into the linguistic characteristics of local government work reports in China.”. Please review this big claim, and explain better what are that ‘practical implications’ and what suppose that for government reports in China.

Introduction

2.- The introduction references Zipf’s seminal works but fails to include more recent and critical discussions on the principle of least effort, optimal coding, information theory, two regimes in Zipf’s law and other relevant linguistic laws not studied here (Heaps-Herdan law, brevity law, Menzerath’s law...). In particular, some basic recent works are missed (see references), works that offer significant insights into the physical and mathematical origins of Zipf’s law that are highly relevant to the study (Debowski, 2020). These references should be included and discussed to establish a stronger theoretical foundation in this introduction, but not only this: other relevant linguistic laws should be analyzed in order to improve this work (i.e. see Torre et al., 2019).

Suggested References:

Debowski, L. (2020). Information theory meets power laws: Stochastic processes and language models. John Wiley & Sons.

Dębowski, Ł., & Bentz, C. (2020). Information theory and language. Entropy, 22(4), 435. See other topics in: https://www.mdpi.com/1099-4300/22/4/435

Font-Clos, F., Boleda, G., & Corral, A. (2013). A scaling law beyond Zipf's law and its relation to Heaps' law. New Journal of Physics, 15(9), 093033.

Català, N., Baixeries, J., & Hernández-Fernández, A. (2024). Exploring Semanticity for Content and Function Word Distinction in Catalan. Languages, 9(5), 179.

Ferrer-i-Cancho, R., Bentz, C., & Seguin, C. (2022). Optimal coding and the origins of Zipfian laws. Journal of Quantitative Linguistics, 29(2), 165-194.

Kello, C. T., Brown, G. D., Ferrer-i-Cancho, R., Holden, J. G., Linkenkaer-Hansen, K., Rhodes, T., & Van Orden, G. C. (2010). Scaling laws in cognitive sciences. Trends in cognitive sciences, 14(5), 223-232.

Semple, S., Ferrer-i-Cancho, R., & Gustison, M. L. (2022). Linguistic laws in biology. Trends in Ecology & Evolution, 37(1), 53-66.

Ferrer i Cancho, R. (2007). On the universality of Zipf’s law for word frequencies. Exact methods in the study of language and text. To honor Gabriel Altmann, 131-140.

Ferrer i Cancho, R., & Solé, R. V. (2001). Two regimes in the frequency of words and the origins of complex Lexicons: Zipf’s law revisited. Journal of Quantitative Linguistics, 8(3), 165-173.

Piantadosi, S. T. (2014). Zipf’s word frequency law in natural language: A critical review and future directions. Psychonomic bulletin & review, 21, 1112-1130.

Torre IG, Luque B, Lacasa L, Kello CT, Hernández-Fernández A. (2019). On the physical origin of linguistic laws and lognormality in speech. R. Soc. Open Sci. 6, 191023.

Ferrer i Cancho, R. (2005). The variation of Zipf’s law in human language. The European Physical Journal B-Condensed Matter and Complex Systems, 44, 249-257.

3.- While the introduction emphasizes the structured and formal nature of LGWRs, it does not sufficiently connect these features to the theoretical implications of Zipf’s law. The authors should discuss how document structure, specialized terminology, and political jargon might influence word frequency distributions. Incorporating this analysis, as well as the sttudy of other linguistic laws like Zipf’s law of brevity, would deepen the reader’s understanding of how linguistic patterns in LGWRs align or deviate from Zipf’s law.

4.- The authors should also discuss the distinction between Zipf’s rank-frequency law and other related linguistic laws, that should be studied (see next points), like Zipf’s law of abbreviation: are the data studied following Zipf’s law of abbreviation? The two zipfian laws are sometimes conflated in the literature. A clearer delineation of these concepts would strengthen the theoretical framework. Specifically, please review if a two regime in Zipf’s law is found in the data (Ferrer i Cancho and Solé, 2001).

REF:

Ferrer i Cancho, R., & Solé, R. V. (2001). Two regimes in the frequency of words and the origins of complex Lexicons: Zipf’s law revisited. Journal of Quantitative Linguistics, 8(3), 165-173.

Literature review

5.- See point 2. A good literature review should include a general review on linguistic laws, and an extended review of Zipf’s law literature (see recent references in point 2). This is a key point of that review. For example, the works by Williams and cols. (2015) are discussed but not other related, comparing linguistic laws in speech and text (Piantadosi, 2014; Torre et al., 2019). Zipfian laws also include meaning-frequency laws, neither studied nor discussed here (see Català et al., 2024, and Català et al., 2021 and references therein).

REFS:

Piantadosi, S. T. (2014). Zipf’s word frequency law in natural language: A critical review and future directions. Psychonomic bulletin & review, 21, 1112-1130.

Català, N., Baixeries, J., Ferrer-i-Cancho, R., Padró, L., & Hernández-Fernández, A. (2021). Zipf’s laws of meaning in Catalan. PloS one, 16(12), e0260849.

Català, N., Baixeries, J., & Hernández-Fernández, A. (2024). Exploring Semanticity for Content and Function Word Distinction in Catalan. Languages, 9(5), 179.

Torre IG, Luque B, Lacasa L, Kello CT, Hernández-Fernández A. (2019). On the physical origin of linguistic laws and lognormality in speech. R. Soc. Open Sci. 6, 191023.

6.- Authors claim that (lines 100-101): “Building on this research, the present study aims to further investigate the applicability of Zipf’s law within the context of Chinese local government work reports” but it is not clear what are the connection between the statistical work done and the suggestion of Chinese local government reports improvement. Please explain it better.

Methods

7.- The methodological section lacks detail on how the Jieba segmentation algorithm was fine-tuned for LGWRs, which likely include specialized political terminology. The authors should describe any preprocessing steps to ensure accurate segmentation and whether manual corrections were made.

8.- In fact, it would be crucial to compare the analysis presented in this text with similar approaches applied to text on the same size (Corral et al., 2015) in other genres, such as literary works, scientific texts, or technical articles. This exercise could reveal common patterns, methodological differences, or unique perspectives that each type of content brings to the analysis of political texts. Furthermore, it would allow for an evaluation of how the tools or theoretical frameworks used in this work could be adapted or complemented in other discursive contexts, thereby fostering a broader and more multidisciplinary perspective. Could you explore these connections and share your insights?

REF:

Corral, Á., Boleda, G., & Ferrer-i-Cancho, R. (2015). Zipf’s law for word frequencies: Word forms versus lemmas in long texts. PloS one, 10(7), e0129031.

9.- The choice of regression models could be appropriate but it is not adequately justified. The authors should explain why Stata’s regression methods were preferred over alternative statistical tools or machine learning approaches that might capture nonlinearities in the data, as well as other approaches (See Zipf’s section in Torre et al., 2019, and the supplementary data code). Why is not studied a double regime? (Ferrer I Cancho and Solé, 2001)?

REF:

Ferrer i Cancho, R., & Solé, R. V. (2001). Two regimes in the frequency of words and the origins of complex Lexicons: Zipf’s law revisited. Journal of Quantitative Linguistics, 8(3), 165-173.

Torre IG, Luque B, Lacasa L, Kello CT, Hernández-Fernández A. (2019). On the physical origin of linguistic laws and lognormality in speech. R. Soc. Open Sci. 6, 191023.

Results

10.- The results section provides a broad overview but lacks depth in exploring the observed heterogeneity in Pareto coefficients. Table 1 should be better explained. The authors should analyze whether these variations correlate with specific historical, regional, or policy-driven factors. For instance, do certain political or economic shifts influence the linguistic patterns in LGWRs? This should be improved studying other text sources (literary, scientific…) of the same size and comparing with the texts studied.

11.- The authors note that LGWRs conform to Zipf’s law but do not critically examine the deviations and possible regimes (Ferrer i Cancho and Solé, 2001). Are these deviations systematic, and if so, what might they reveal about the dynamics of political communication? Are the meanings of the words related with frequency? With an analysis of the data using a Chinese dictionary (to obtain the number of meanings), authors can test the Zipf’s meaning-frequency law, to see if governement texts are more easy to understand that other kind of corpus (comparing oral corpus and written corpus, see Català et al., 2021).

REF:

Ferrer i Cancho, R., & Solé, R. V. (2001). Two regimes in the frequency of words and the origins of complex Lexicons: Zipf’s law revisited. Journal of Quantitative Linguistics, 8(3), 165-173.

Català, N., Baixeries, J., Ferrer-i-Cancho, R., Padró, L., & Hernández-Fernández, A. (2021). Zipf’s laws of meaning in Catalan. PloS one, 16(12), e0260849.

Discussion

12.-The discussion overgeneralizes the implications of the study, claiming that adherence to Zipf’s law can enhance report clarity and transparency. While this is an interesting hypothesis, it requires empirical validation. The authors should either provide evidence supporting their claims or frame it as a potential avenue for future research, comparing with other types of data.

13.- The discussion could be enriched by addressing alternative explanations for the findings. For instance, how might the hierarchical structure of Chinese governance or regional linguistic variations influence the observed patterns?

14.- The authors should consider whether their findings align with general principles in quantitative linguistics (Debowsky, 2020; Baixeries et al., 2013), and how this principle interacts with the specific context of political discourse.

REFS:

Baixeries, J., Elvevåg, B., & Ferrer-i-Cancho, R. (2013). The evolution of the exponent of Zipf's law in language ontogeny. PloS one, 8(3), e53227.

Debowski, L. (2020). Information theory meets power laws: Stochastic processes and language models. John Wiley & Sons.

Conclusion

15. The conclusion is clear but overly simplistic. The authors should briefly reiterate the study’s contributions to both quantitative linguistics and political communication while acknowledging its limitations. Future research directions, such as comparative studies with other countries’ government reports or different kinds of texts/corpora, should be suggested.

16. Final comment. The authors can be pioneers in this type of analysis of linguistic laws in political discourse, which is why I invite a deep revision of this article and its subsequent re-submission. Congratulations and thank you for studying linguistic laws in this interdisciplinary way.

Sincerely yours,

The reviewer

6. PLOS authors have the option to publish the peer review history of their article (what does this mean? ). If published, this will include your full peer review and any attached files.

**Do you want your identity to be public for this peer review?** For information about this choice, including consent withdrawal, please see our Privacy Policy .

Reviewer #1: No

Reviewer #2: No

---

## [Author Response · Author response to Decision Letter 0]

27 Dec 2024

Dear Dr. Pruller and Reviewers,

Thank you for your insightful comments and suggestions on our manuscript, “Zipf’s Law in China’s Local Government Work Reports: A 21-Year Study Using Natural Language Processing and Regression Analysis”. We really appreciate the time and effort dedicated to evaluating the work and find the feedback invaluable in strengthening the manuscript. We have carefully addressed each point raised and believe the revisions significantly enhance the study’s rigor, clarity, and overall impact. All changes have been highlighted in red in the manuscript in response to the reviewers’ comments, and a list of the main modifications include:

•We have revised the abstract, introduction and literature review.

•We have refined the methodological details by providing specific information on the Jieba word segmentation tool, the implementation of a double-logarithmic model, and the development of a custom dictionary. Furthermore, we have enhanced the analysis of our findings, with a particular emphasis on the substantial inter-regional and inter-temporal variations observed in the Pareto coefficients.

•We have included detailed descriptive statistics of the length of government work reports,which is now in table 1.

•We have made the limitations and future research directions more concrete in the conclusion section. Additionally, we have removed unsubstantiated claims about practical implications.

In the subsequent sections, we have included a detailed response to the feedback from both reviewers. The reviewers’ comments are marked in blue, while our replies are presented in black.

Sincerely,

Yanfang LI

---

## [Decision Letter · Decision Letter 1]

22 Jan 2025

PONE-D-24-44352R1Zipf's law in China's local government work reports: A 21-year study using natural language processing and regression analysisPLOS ONE

Dear Dr. Li,

Thank you for submitting your manuscript to PLOS ONE. After careful consideration, we feel that it has merit but does not fully meet PLOS ONE’s publication criteria as it currently stands. Therefore, we invite you to submit a revised version of the manuscript that addresses the points raised during the review process.

**Dear [Author(s)],**

I am writing regarding your manuscript, *"Zipf’s Law in China’s Local Government Work Reports: A 21-year Study Using Natural Language Processing and Regression Analysis,"* submitted to *PLOS ONE* . As the original handling editor was unavailable to continue overseeing your manuscript, I have stepped in to process the submission at the revision stage.

Of the two original reviewers, only one was able to assess the revised manuscript, and they expressed satisfaction with the changes made. Unfortunately, despite repeated requests, the second reviewer, who had provided the bulk of the comments during the first round, could not be reached. To ensure a thorough evaluation, I invited two new reviewers to assess your revised submission. Both have provided excellent feedback and have recommended major revisions, providing thoughtful and constructive comments and suggestions.

I want to express my appreciation for your diligent work in addressing the feedback from the initial round of reviews. Based on the reviews and my in-depth reading of the manuscript, I believe the paper is making significant progress toward eventual publication. I have compiled a comprehensive list of actionable items drawn from the reviewers’ comments and my own observations. These suggestions are intended to strengthen the manuscript further and ensure it meets the high standards of *PLOS ONE* .

I trust that you will approach these recommendations positively and provide a revised manuscript addressing the points outlined below.

**Comprehensive List of Action Items**

**Analysis Suggestions**

**Regional Differences Need Better Storytelling:**As noted by reviewers, it is hard to discern regional differences from the current presentation. To address this:Plot the **trend of the top 20%** (e.g., provinces/mandates with the highest Zipf coefficients) and **contrast it with the bottom 20%.**Perhaps the differences are between rural and urban regions? (As an example of a paper that tells a visual story nicely in *PLOS ONE* : Shankar, S.G., Miller, J.M., and Balakrishnan, P.V. (2020), "Evolutionary disruption of S&P 500 trading concentration: An intriguing tale of a financial innovation," *PLOS ONE,* 15(3), p.e0230393, is a potentially useful template for describing and studying evolutionary changes in the power law coefficient.)To this end, **improve Figure 2 representation** :The current Figure 2 is a blur of dots and does not effectively convey meaningful insights. Consider an alternative representation that highlights trends or patterns more clearly.
Use such comparisons to tell a more compelling story. For example:Are provinces/mandates evolving in lockstep?Are Zipf coefficients for documents from different regions evolving differently?Or are all changes seemingly random?
**Comparison with Consequential Paper:**How do these findings compare with the statement by **Naranan & Balasubrahmanyan (1998)** ?

*"A word frequency analysis of a text can reveal nothing about its characteristics (author, language, style, type of literature, etc.). The only exception appears to be Shakespeare with ζ = 1.6; this fact has indeed been exploited in attempts to resolve some controversies about authorship of certain texts (Thisted & Efron, 1987)."*

**Feedback** : Discuss whether your findings align with or challenge this claim in your context, particularly regarding whether regional or stylistic differences are discernible in Zipf coefficients.

**Motivation and Contribution**

**Clarify the Contribution of the Study:**Clearly articulate how this study advances knowledge beyond existing research on Zipf's law in Chinese language and documents.Revise the last sentence on Page 3, Lines 55-59 (P3.55-59), to reflect the actual contribution of the study instead of repeating the dataset and aim.Differentiate this study by highlighting your novel aspects (e.g., unique dataset, focus on local government work reports, or insights into political communication).Incorporate findings from regional trends and comparison with prior research to show unique insights into regional or stylistic differences in political communication and linguistic evolution.
**Elaborate on Contributions:**Emphasize how findings contribute to understanding regional linguistic styles or political communication, if you find any after your revised analysis.Highlight implications of differences among provinces/regions beyond overall conformity to Zipf’s law.

**Specific Minor Comments**

**Typo in Line 220:**Correct "approximately1." to "approximately 1." (space missing).
**Delete Lines 223-224:**Remove: "While machine learning methods excel at predicting future trends, this study focuses on historical data and the relationships between variables." **Reason** : Adds nothing to the overall context.
**Claim on Universal Applicability:**Reword "demonstrating its universal applicability" to "demonstrating its potential universal applicability in Indian languages" (P3.64-66).
**Definition of Chinese Language and Scope:**Clarify whether Tibetan is treated as a "Chinese language" or a distinct language used in China (P4.74-75).Define what "Chinese language" encompasses (e.g., Mandarin, regional dialects, languages spoken within the People's Republic of China).
**IMPORTANT — Delete Politically Charged Aspects:**Delete mention of Taiwan (P6.117-118) to avoid controversial statements in a non-political academic paper. Best is to delete names of all provinces you have NOT studied. (see Reviewer #4 as well)
**Inconsistencies in Pareto Coefficient Definitions:**Clarify (see Reviewer #4) whether the Pareto coefficient is represented by lnC or β in the manuscript (P7.141 & P8.165).Explain the relationship between the calculated β values in regressions and the theoretical value of the Pareto coefficient (β = 1).Address why the maximum β values reported in Table 1 are significantly below 1 and reconcile this with the conclusion that Chinese local government work reports conform to Zipf’s law.
**Decimal Places in Lines 260-261:**Reduce excessive decimal places in the coefficient range (e.g., "0.6678291 to 0.9252278") to a more reasonable number (e.g., 2 or 3).
**Overclaim in Line 288:**Avoid overstating conclusions by revising: "indicating that while most reports conform to Zipf’s law."
**Engage with Suggested Literature:**Reference the suggested article (https://www.colips.org/journals/volume18/JCLC_2008_V18_N1_04.pdf ) and clarify how this study complements, differs from, or builds upon it.

I look forward to receiving your revised manuscript. Should you have any questions or require clarification on any of the points above, please do not hesitate to contact me.

Sincerely,

Academic Editor, *PLOS ONE*

We look forward to receiving your revised manuscript.

Kind regards,

P. V. (Sundar) Balakrishnan, Ph.D

Academic Editor

PLOS ONE

Reviewers' comments:

Reviewer's Responses to Questions

**Comments to the Author**

1. If the authors have adequately addressed your comments raised in a previous round of review and you feel that this manuscript is now acceptable for publication, you may indicate that here to bypass the “Comments to the Author” section, enter your conflict of interest statement in the “Confidential to Editor” section, and submit your "Accept" recommendation.

Reviewer #2: All comments have been addressed

Reviewer #3: (No Response)

Reviewer #4: (No Response)

2. Is the manuscript technically sound, and do the data support the conclusions?

Reviewer #2: Yes

Reviewer #3: (No Response)

Reviewer #4: (No Response)

3. Has the statistical analysis been performed appropriately and rigorously? 

Reviewer #2: Yes

Reviewer #3: (No Response)

Reviewer #4: (No Response)

4. Have the authors made all data underlying the findings in their manuscript fully available?

Reviewer #2: Yes

Reviewer #3: (No Response)

Reviewer #4: (No Response)

5. Is the manuscript presented in an intelligible fashion and written in standard English?

Reviewer #2: Yes

Reviewer #3: (No Response)

Reviewer #4: (No Response)

6. Review Comments to the Author

Reviewer #2: The authors have taken into account almost all of my comments, eliminating some categorical statements or those referring to the improvement of communication by the Chinese government (something that cannot be proven), so I consider that the modifications make the article publishable. I only make some final notes and typos for improvement:

- Lines 95-97. Authors claim that " This investigation contributes to a deeper understanding of the applicability of Zipf’s law within the context of formalized political communication." Please clarify shortly how, in what way quantitative linguistics can do that, to explain this to a broader audience.

- Line 116. Torre et al. ->Torre et al. (2019). Please review all the manuscript to add year of publication when quoting direct name of author's paper.

- Line 138. Authors claim "In summary, whereas understanding of Zipf’s law was initially simplistic,..." I think it is better "In summary, whereas Zipf’s law was mathematically simple,..." because maths proposed by Zipf are easy but not deep concepts, in my opinion.

- Line 141. Menzerat’s -> Menzerath’s

- Section 3.2. please clarify how the transition point between two zipfian regime is found.

Finally, I encourage the authors to continue studying other linguistic laws in the materials they have (as they suggest they will do in the future work), as well as to analyze the semantics of political communications, something very important socially.

Reviewer #3: The paper was originally reviewed by two reviewers and the author has responded to those comments. I think the referee has responded to those comments reasonably well.

However, the main issue is the contribution of this study. There are so many articles that have examined the relevance of Zipf's law in the context of Chinese language; and generally report that Chinese language/ reports follow Zipf's law. Therefore, I am not sure about the contribution of this study. It looks a bit mechanical. For example, please see:

https://www.colips.org/journals/volume18/JCLC_2008_V18_N1_04.pdf

The readers should be convinced about the contribution of this study - given so many existing studies on Chinese document in the context of Zipf's law! How does this study improve our knowledge?

Reviewer #4: (No Response)

7. PLOS authors have the option to publish the peer review history of their article (what does this mean? ). If published, this will include your full peer review and any attached files.

**Do you want your identity to be public for this peer review?** For information about this choice, including consent withdrawal, please see our Privacy Policy .

Reviewer #2: No

Reviewer #3: No

Reviewer #4: No

---

## [Author Response · Author response to Decision Letter 1]

9 Feb 2025

Dear editor and reviewer,

Thank you for the opportunity to revise our manuscript. We have carefully addressed all editor and reviewer comments and believe the revised version is significantly improved. We are happy to provide any further information if needed.

---

## [Decision Letter · Decision Letter 2]

25 Feb 2025

PONE-D-24-44352R2Zipf's law in China's local government work reports: A 21-year study using natural language processing and regression analysisPLOS ONE

Dear Dr. Li,

Thank you for submitting your manuscript to PLOS ONE. After careful consideration, we feel that it has merit but does not fully meet PLOS ONE’s publication criteria as it currently stands. Therefore, we invite you to submit a revised version of the manuscript that addresses the points raised during the review process.

Dear Dr. Li,

I have now spent a considerable amount of time reading over and trying to replicate our analyses and to ensure compliance with the reviewers' comments as well as that the manuscript meets the guidelines for publication in PLOS ONE. I want to thank the reviewers for their extraordinary timeliness and their excellent feedback over the last two rounds. I am inviting you to undertake a through revision of your manuscript. It is eminently feasible to do so, and I hope you will work to improve it. This is a major revision, and I don't want to drag it on. I would like to make the decision to accept or reject in the next round. The FIRST ISSUE is the reporting of the values of the coefficients.

**Rank-Frequency Law in Word Distributions (Zipf’s Law):** The **Rank-Frequency Law** , also known as **Zipf’s Law** , describes the relationship between the **frequency** of a word in a corpus and its **rank** in a descending ordered list of word occurrences. The **Rank-Size Law** as you have correctly stated in equation (on Line 226) states that the frequency (<img height="31" src="file:///C:/Users/sundar/AppData/Local/Temp/msohtmlclip1/01/clip_image002.png" width="9" />) of a word is inversely proportional to its rank (<img height="31" src="file:///C:/Users/sundar/AppData/Local/Temp/msohtmlclip1/01/clip_image004.png" width="11" />) raised to a power **α** :

**SEE ATTACHMENT FOR FORMULAE AND DERIVATION ETC (As the formatting does not paste well here) **

But it seems like the reviewer has pointed out, this reciprocal transformation may have not been implemented. In either case, your setup and writing of this aspect is rather confusing as you move from Zipf’s Law to Pareto Distribution etc. I suggest that given the focus of your paper is on ZIPF’s LAW as you have used in the title of your manuscript, you stick with rporting and analyzing and plotting the Zipf Coefficients. For the connection between Zipf and Pareto please see the note by Lada A. Adamic. This tutorial explains the relationship between Zipf’s law, power-law distributions, and Pareto’s law, highlighting how they describe similar phenomena in different ways.

**Source:**
https://web.cs.dal.ca/~shepherd/courses/csci4141/zipf/ranking.html
**Zipf, Power-laws, and Pareto - a ranking tutorial, Lada A. Adamic.**   This, a higher Zipf coefficient corresponds to a more uneven distribution with a lower Pareto coefficient; Conversely, a lower Zipf coefficient reflects a more even distribution with a higher Pareto coefficient.

**This leads to an easier interpretation of the Zipf Coefficient (α).**

**If α ≈ 1** , the data follows **Zipf’s Law** , meaning word frequencies follow a typical rank-size rule.**If α > 1** , the distribution is **steeper** , meaning a few high-frequency words dominate the text.**If α < 1** , the distribution is **flatter** , meaning word frequencies are more evenly distributed.

I recommend that you rewrite this so we have a clean paper with respect to Zipf’s Law.

SECOND ASPECT AS THE REVIEWER HAS POINTD OUT IS THE CONTRIBUTION. This can be rectified as follows. **Zipf Coefficients movements Need Better Storytelling:**

As noted by reviewers, it is not iclear as to what your findings are. At this stage, I am willing to suggest that the major contribution hs been to conduct a challenging and exahustive analysis to it is hard to discern regional differences from the current presentation. To address this:Plot the **trend of the top (10%) 20%** with the highest Zipf coefficients and **contrast it with the bottom (10%) 20%. I conducted this analysis (based on the limited data you have made available of the coefficeints in excel file).** I present the plot below. There is a significant divergence in the two plots. I am also able to see a natural break-point from this plot at about year 2011 or so. I used as template a paper in *PLOS ONE* : Shankar, S.G., Miller, J.M., and Balakrishnan, P.V. (2020), “Evolutionary disruption of S&P 500 trading concentration: An intriguing tale of a financial innovation,” *PLOS ONE,* 15(3), p.e0230393, which was very useful for describing and studying evolutionary changes in the Zipf coefficient.Please do this carefully, but using the ZIPF Coefficients, so that the connection can be made easier to the Principle of Least Effort.Perhaps the differences are between rural and urban regions? Please indicate in your table with one identifier as to which are Rural and which are Urban.Unfortunately, Figures 2 to 4 are nothing but a *blur of blue dots and do not effectively convey meaningful insights* . Consider an alternative representation that highlights trends or patterns more clearly. Please replace them with the suggested plots. Use color coding to make them tell a visual story. Use such comparisons to tell a more compelling story. For example, I APPEND A SAMPLE OF A PLOT I WAS ABLE TO CONSTRUCT USING YOUR ESTIMATES, BUT THESE ALL HAVE TO BE REWORKED TO USING THE CORRECTED VALUES:

<h3>**Other Recommendations for Enhancing the Analysis and Presentation** </h3>

**1. Strengthening the Visual Narrative**

the authors should **plot the top 20% and bottom 20% of Zipf values**  across all regions for each year; **comparison of the 10% and 20% groupings**  in a single plot would allow for better trend visualization. Overlaying both **10% and 20% plots**  on the same figure will help in assessing the stability of rankings. In addition, I was able to identify the natural brak point which seems to correspond with the 18th National Congress of the Communist Party of China in 2012.

**2. Identifying Key Regions in Extreme Categories**

The authors should **determine the specific regions that consistently appear in the top 10% (20%) and bottom 10%**  (20%) of Zipf values over time. A **table**  should be included to highlight persistent high- and low-ranking regions.

<h4>**3. Regional Movement and Stability Analysis** </h4>

Additionally, it would be useful to   identify which regions frequently transition between these groups. If possible see if there are some geographical or political aspects as to why this occurs. A **table summarizing how many times each region appeared in the top 20% or bottom 20% over the years**  should be included. Another set should identify   **regions that never appeared in either extreme** , as this suggests stability.**Persistence and Volatility Metrics:**  The authors should   **calculate persistence scores**   to rank regions by how consistently they appear in either the top or bottom 20%. Conversely, the authors should identify   **regions that frequently switch between top and bottom groups**   to highlight volatility. The discussion should explore   **why some regions are stable while others are more dynamic**   in their rankings.

THIRD, AND THIS IS CRUCIAL. PLOS Data policy <o:p></o:p>

A big hurdle is the the authors have NOT made all data underlying the findings in their manuscript fully available. The PLOS Data policy  requires authors to make all data underlying the findings described in their manuscript fully available without restriction, with rare exception I was able to access ONLY their reported estimate of the Pareto Coefficients. It was not possible for me to replicate or help them enhance their analysis. The only folder to a region is Beijing and that is empty. **Please make sure to upload the corpus frequencies data.**

<o:p></o:p>

STYLISTIC AND SMALLER EDITS / COMMENTS:

Lines 229-230 are confusing the metrics and can be eliminated.

Line 240. Reference to “Torrre et al (2019) [13]” is strange, pick one citation reference style.

Line 337 - 340: DELETE “Our findings directly challenge the assertion by Naranan and Balasubrahmanyan (1998) that word frequency analysis reveals nothing about a text’s characteristics (e.g., author, language, style, and type of literature) [37].” Currently, the paper has not made a substantive contribution on these matters. At best you have some suggestive findings that need investigation.

STATA MLE ESTIMATION: Please clearly specify as to what package(s) were used to estimate the coefficients. Are you using “ml maximize”? How did you define the likelihood function? Sample code use for the MLE should be provided to ensure replication.

In Table 1 Descriptive statistics on the length of government work reports, please clarify the units. I assume that these are number of unique “words” in each corpus-year.

I think you have a very good chance for an excellent paper. Please do get this done as it is not very difficult, but is a lot of painstaking work. Thank you for your pateience.   

We look forward to receiving your revised manuscript.

Kind regards,

Sundar P. V. Balakrishnan, Ph.D

Academic Editor

PLOS ONE

Reviewers' comments:

Reviewer's Responses to Questions

**Comments to the Author**

1. If the authors have adequately addressed your comments raised in a previous round of review and you feel that this manuscript is now acceptable for publication, you may indicate that here to bypass the “Comments to the Author” section, enter your conflict of interest statement in the “Confidential to Editor” section, and submit your "Accept" recommendation.

Reviewer #3: All comments have been addressed

Reviewer #4: (No Response)

2. Is the manuscript technically sound, and do the data support the conclusions?

Reviewer #3: Yes

Reviewer #4: (No Response)

3. Has the statistical analysis been performed appropriately and rigorously? 

Reviewer #3: Yes

Reviewer #4: (No Response)

4. Have the authors made all data underlying the findings in their manuscript fully available?

Reviewer #3: Yes

Reviewer #4: (No Response)

5. Is the manuscript presented in an intelligible fashion and written in standard English?

Reviewer #3: Yes

Reviewer #4: (No Response)

6. Review Comments to the Author

Reviewer #3: The author has highlighted the contributions in the revived version. I am okay with it, although I feel that the contributions are modest.

Reviewer #4: (No Response)

7. PLOS authors have the option to publish the peer review history of their article (what does this mean? ). If published, this will include your full peer review and any attached files.

**Do you want your identity to be public for this peer review?** For information about this choice, including consent withdrawal, please see our Privacy Policy .

Reviewer #3: No

Reviewer #4: No

---

## [Author Response · Author response to Decision Letter 2]

21 Mar 2025

dear editor and reviewer,

Thank you for your thoughtful comments and suggestions regarding our manuscript. We truly appreciate the time and effort you have invested in reviewing our work, and your feedback has been extremely valuable in improving the manuscript. We have carefully considered each of your points and made revisions that we believe greatly enhance the study’s rigor, clarity, and overall quality.

---

## [Editor Report · Decision Letter 3]

26 Mar 2025

PONE-D-24-44352R3Zipf's law in China's local government work reports: A 21-year study using natural language processing and regression analysisPLOS ONE

Dear Dr. Li,

Thank you for submitting your manuscript to PLOS ONE. After careful consideration, we feel that it has merit but does not fully meet PLOS ONE’s publication criteria as it currently stands. Therefore, we invite you to submit a revised version of the manuscript that addresses the points raised during the review process.

We look forward to receiving your revised manuscript.

Kind regards,

P. V. (Sundar) Balakrishnan, Ph.D

Academic Editor

PLOS ONE

Journal Requirements:

Additional Editor Comments :

The Author has been very responsive and made significant efforts to address the comments and excellent suggestions made in the review process. The manuscript is significantly improved in terms of its contributions and correctness.

1. The data is now available as per my last check

The interpretability of the results is enhanced and the figures do a nice job are no longer just a blur of blue dots!

2. Some small corrections/ edits should be made:

What were the STATA commands used - please specify clearly?

" = frequency of the word in the corpus" Did you use raw count of words; or the percentage?

Line 356: DELETE "contradicting Naranan and Balasubrahmanyan’s claim [37]." unnecessarily contentious as I am not sure we can go into the claims and debae with teh limited information in this paper.

Table 2: use only appropriate number of decimals - to make it easier for the readers.
---

## [Author Response · Author response to Decision Letter 3]

7 Apr 2025

Dear reviewer and editor,

Thank you for the constructive feedback from you and the editors. We have carefully addressed all comments and revised the manuscript accordingly. Please let us know if any further modifications are needed. We appreciate your time and look forward to your decision.

---

## [Editor Report · Decision Letter 4]

10 Apr 2025

PONE-D-24-44352R4Zipf's law in China's local government work reports: A 21-year study using natural language processing and regression analysisPLOS ONE

Dear Dr. Li,

Thank you for submitting your manuscript to PLOS ONE. After careful consideration, we feel that it has merit but does not fully meet PLOS ONE’s publication criteria as it currently stands. Therefore, we invite you to submit a revised version of the manuscript that addresses the points raised during the review process.

Thank you for your submission. It is close to publication ready. But I see a couple of bad errors that you should correct before it is published. These are clarification regarding the regression model presented in lines 249–254, particularly your implementation of Zipf’s Law.

**Lognormal vs. Power-Law Terminology**

The phrase “based on the analysis of lognormal distributions” is misleading. Zipf’s Law implies a **power-law** distribution, not a **lognormal** .

**Regression Specification – Estimated vs. Fixed Slope**

Your current equation:

log(S_it) = log(c) - log(R_it) + ε

is not what I suspect you estimate! As implies a fixed slope of –1, which assumes perfect adherence to Zipf’s Law. If the goal is to test Zipf’s Law empirically, I would recommend estimating the slope from the data instead of fixing it. The more general and commonly used formulation is:

log(S_it) = β_0 + β_1 * log(R_it) + ε

Then, you can test whether β_1 ≈ –1 to assess the degree of adherence to Zipf’s Law.

**Suggested Revision for Lines 249–254:**

“To empirically verify Zipf’s Law, we estimate a double-logarithmic regression of the form:

log(S_it) = β_0 + β_1 * log(R_it) + ε

where S_it denotes the observed size (e.g., word frequency), R_it is the rank of the item in the corpus for  IMPORTANT: Please read the paper over to double check for any and all small errors and edits.  

We look forward to receiving your revised manuscript.

Kind regards,

P. V. Sundar Balakrishnan, Ph.D.

Academic Editor

PLOS ONE
---

## [Author Response · Author response to Decision Letter 4]

24 Apr 2025

Dear Dr. Sundar Balakrishnan,

Thank you for your thoughtful feedback. We sincerely appreciate your time and expertise in reviewing our manuscript. We have carefully addressed all comments and revised the manuscript accordingly. Additionally, we have meticulously proofread the manuscript multiple times to correct minor grammatical, spelling, and formatting inconsistencies.

---

## [Editor Report · Decision Letter 5]

28 Apr 2025

PONE-D-24-44352R5Zipf's law in China's local government work reports: A 21-year study using natural language processing and regression analysisPLOS ONE

Dear Dr. Li,

Thank you for submitting your manuscript to PLOS ONE. After careful consideration, we feel that it has merit but does not fully meet PLOS ONE’s publication criteria as it currently stands. Therefore, we invite you to submit a revised version of the manuscript that addresses the points raised during the review process.

Thank you for your careful and thorough work. Overall, the manuscript is now significantly better and clearer. However, please address the following minor issues for clarity and consistency:

**Line 249 Correction:**Currently: " = 1,2 … 31 ,t"Correction Needed: Insert a semicolon after "31" and add a space before "t": " = 1,2 … 31; t"
**Lines 307–308 Restatement:**Current statement: "...only 1% of the coefficients fall below 0.722, while 10% are below 0.791. This suggests that the content distribution in most local government work reports exhibits a high degree of concentration, ..."Suggested revision: Clarify this statement in alignment with your definition provided in lines 234–237. Given your definition that a coefficient (β) greater than 1 indicates a more skewed distribution dominated by high-frequency words, and a coefficient less than 1 indicates a flatter, more even distribution, it would be clearer to explicitly state that none of the coefficients exceed 1, indicating overall a relatively even distribution of word frequencies.

Once these minor issues are addressed, your manuscript can be accepted. Though it might be useful to check the printed version carefully.

We look forward to receiving your revised manuscript.

Kind regards,

Sundar P. V. Balakrishnan, Ph.D

Academic Editor

PLOS ONE
---

## [Author Response · Author response to Decision Letter 5]

29 Apr 2025

Dear editor,

Thank you for your valuable feedback. We have carefully implemented the suggested revisions. We confirm all changes have been made in the manuscript. Please let us know if further adjustments are needed.

---

## [Editor Report · Decision Letter 6]

30 Apr 2025

Zipf's law in China's local government work reports: A 21-year study using natural language processing and regression analysis

PONE-D-24-44352R6

Dear Dr. Li,

We’re pleased to inform you that your manuscript has been judged scientifically suitable for publication and will be formally accepted for publication once it meets all outstanding technical requirements.

Kind regards,

Sundar P. V. Balakrishnan, Ph.D

Academic Editor

PLOS ONE
---

## [Editor Report · Acceptance letter]

PONE-D-24-44352R6

PLOS ONE

Dear Dr. Li,

I'm pleased to inform you that your manuscript has been deemed suitable for publication in PLOS ONE. Congratulations! Your manuscript is now being handed over to our production team.

Kind regards,

on behalf of

Dr. Sundar P. V. Balakrishnan

Academic Editor

PLOS ONE